# Reconciling kinetic and thermodynamic models of bacterial transcription

**Muir Morrison**[1], **Manuel Razo-Mejia**[2], **Rob Phillips**[1,2]*

**1** Department of Physics, California Institute of Technology, Pasadena, California, USA, **2** Division of Biology and Biological Engineering, California Institute of Technology, Pasadena, California, USA

* phillips@pboc.caltech.edu

## Abstract

The study of transcription remains one of the centerpieces of modern biology with implications in settings from development to metabolism to evolution to disease. Precision measurements using a host of different techniques including fluorescence and sequencing readouts have raised the bar for what it means to quantitatively understand transcriptional regulation. In particular our understanding of the simplest genetic circuit is sufficiently refined both experimentally and theoretically that it has become possible to carefully discriminate between different conceptual pictures of how this regulatory system works. This regulatory motif, originally posited by Jacob and Monod in the 1960s, consists of a single transcriptional repressor binding to a promoter site and inhibiting transcription. In this paper, we show how seven distinct models of this so-called simple-repression motif, based both on thermodynamic and kinetic thinking, can be used to derive the predicted levels of gene expression and shed light on the often surprising past success of the thermodynamic models. These different models are then invoked to confront a variety of different data on mean, variance and full gene expression distributions, illustrating the extent to which such models can and cannot be distinguished, and suggesting a two-state model with a distribution of burst sizes as the most potent of the seven for describing the simple-repression motif.

## Author summary

With the advent of new technologies allowing us to query biological activity with ever increasing precision, the deluge of quantitative biological data demands quantitative models. Transcriptional regulation—a feature that lies at the core of our understanding of cellular control in myriad context ranging from development to disease—is no exception, with single-cell and single-molecule techniques being routinely deployed to study cellular decision making. These data have served as a fertile proving ground to test models of transcription that mainly come in two flavors: thermodynamic models (based on equilibrium statistical mechanics) and kinetic models (based on chemical kinetics). In this paper we study the correspondence between these theoretical frameworks in the context of the simple repression motif, a common regulatory architecture in prokaryotes in which a repressor with a single binding site regulates expression. We explore the consequences of

**Data Availability Statement:** All data and custom scripts were collected and stored using Git version control. Code for Bayesian inference and figure generation is available on the GitHub repository

(https://github.com/RPGroup-PBoC/bursty_transcription).

**Funding:** This material is based upon work supported by the National Science Foundation Graduate Research Fellowship under Grant No. DGE-1745301 (to M.J.M.). This work was also supported by La Fondation Pierre-Gilles de Gennes, the Rosen Center at Caltech, and the NIH 5R35GM118043-05 (MIRA) to R.P. M.R.M. was supported by the Caldwell CEMI fellowship. The funders had no role in study design, data collection and analysis, decision to publish, or preparation of the manuscript.

**Competing interests:** The authors have declared that no competing interests exist.

different levels of coarse-graining of the molecular steps involved in transcription, finding that, at the level of mean gene expression, the different models cannot be distinguished. We then study higher moments of the gene expression distribution which allows us to discard several of the models that disagree with experimental data and supporting a minimal kinetic model.

## Introduction

Gene expression presides over much of the most important dynamism of living organisms. The level of expression of batteries of different genes is altered as a result of spatiotemporal cues that integrate chemical, mechanical and other types of signals. As our ability to experimentally observe and measure the dynamical processes that constitute the central dogma improves, there is an opportunity to undertake a theory-experiment dialogue in order to sharpen our understanding of such a fundamental biological process. One of the remaining outstanding challenges to have emerged in the genomic era is our continued inability to predict the regulatory consequences of different regulatory architectures, i.e. the arrangement and affinity of binding sites for transcription factors and RNA polymerases on the DNA. This challenge stems first and foremost from our ignorance about what those architectures even are, with more than 60% of the genes even in an ostensibly well understood organism such as *E. coli* having no regulatory insights at all [1–4]. But even once we have established the identity of key transcription factors and their binding sites for a given promoter architecture, there remains the predictive challenge of understanding its input-output properties, an objective that can be met by a myriad of approaches using the tools of statistical physics [5–24]. One route to such predictive understanding is to focus on the simplest regulatory architecture and to push the theory-experiment dialogue to increase the predictive power of our theoretical models [25, 26]. If we demonstrate that we can pass that test by successfully predicting both the means and variance in gene expression at the mRNA level, then that provides a more solid foundation upon which to launch into more complex problems—for instance, some of the previously unknown architectures uncovered in [2] and [27].

To that end, in this paper we examine a wide variety of distinct models for the simple repression regulatory architecture. This genetic architecture consists of a DNA promoter regulated by a transcriptional repressor that binds to a single binding site as developed in pioneering early work on the quantitative dissection of transcription [28, 29]. One of the main features of the models we explore is that, by construction, all features related to the microstates in which the repressor is bound to the promoter can be separated from the microstates in which the RNA polymerase (RNAP) is bound. From a modeling perspective, this means that some of the models can be written as effective two-state models for which there is a rich literature [17, 21, 24, 30–36]. Here, we systematically compare the predictions of several models with different levels of coarse graining written in terms of thermodynamic and kinetic parameters. One goal in exploring such coarse-grainings is to build towards the future models of regulatory response that will be able to serve the powerful predictive role needed to take synthetic biology from a brilliant exercise in enlightened empiricism to a rational design framework as in any other branch of engineering. More precisely, we want phenomenology in the sense of coarse-graining away atomistic detail, but still retaining biophysical meaning. In particular a key question is: at this level of coarse-graining, what microscopic details do we need to explicitly model, and how do we figure that out? For example, do we need to worry about all or even any of the steps that individual RNA polymerases go through each time they make a transcript?

Turning the question around, can we see any imprint of those processes in the available data? If the answer is no, then those processes are irrelevant for our purposes. Forward modeling and inverse (statistical inferential) modeling are necessary to tackle such questions. We combine both approaches in order to discard models that cannot empirically satisfy the main features of experimental data. First we apply forward modeling to demonstrate that none of the models are distinguishable at the level of mean gene expression. We then extend the modeling to look at higher moments of the distribution, eliminating models that do not empirically satisfy the observed cell-to-cell variability. Finally we arrive at a minimal model on which we can apply inverse modeling in order to infer the parameters that explain the data.

Fig 1A shows the qualitative picture of simple repression that is implicit in the repressor-operator model. An operator, i.e., the binding site on the DNA for a repressor protein, may be found occupied by a repressor, in which case transcription is blocked from occurring. Alternatively, that binding site may be found unoccupied, in which case RNA polymerase (RNAP) may bind and transcription can proceed. The key assumption we make in this simplest incarnation of the repressor-operator model is that binding of repressor and RNAP in the promoter region of interest is exclusive, meaning that one or the other may bind, but never may both be simultaneously bound. It is often imagined that when the repressor is bound to its operator, RNAP is sterically blocked from binding to its promoter sequence. Current evidence suggests this is sometimes, but not always the case, and it remains an interesting open question precisely how a repressor bound far upstream is able to repress transcription [1]. Suggestions include "action-at-a-distance" mediated by kinks in the DNA, formed when the repressor is bound, that prevent RNAP binding. Nevertheless, our modeling in this work is sufficiently coarse-grained that we simply assume exclusive binding and leave explicit accounting of these details out of the problem.

The logic of the remainder of the paper is as follows. In the section 1, we show how both thermodynamic models (Fig 1B) and kinetic models based upon the chemical master equation (Fig 1C) all culminate in the same underlying functional form for the fold-change in the average level of gene expression with an effective free energy $\Delta F_R$ capturing the regulation given by the transcription factor, and a term $\rho$ describing the level of coarse-graining of the transcriptional events as shown in Fig 1D. Section 2 goes beyond an analysis of the mean gene expression by asking how the same models presented in Fig 1C can be used to explore noise in gene expression. To make contact with experiment, all of these models must make a commitment to some numerical values for the key parameters found in each such model. Therefore in Section 3 we explore the use of Bayesian inference to establish these parameters and to rigorously answer the question of how to discriminate between the different models.

## Materials and methods

All data and custom scripts were collected and stored using Git version control. Code for Bayesian inference and figure generation is available on the GitHub repository (https://github.com/RPGroup-PBoC/bursty_transcription).

## Results

### 1 Mean gene expression

As noted in the previous section, there are two broad classes of models in play for computing the input-output functions of regulatory architectures as shown in Fig 1. In both classes of model, the promoter is imagined to exist in a discrete set of states of occupancy, with each such state of occupancy accorded its own rate of transcription–including no transcription for many of these states. This discretization of a potentially continuous number of promoter states

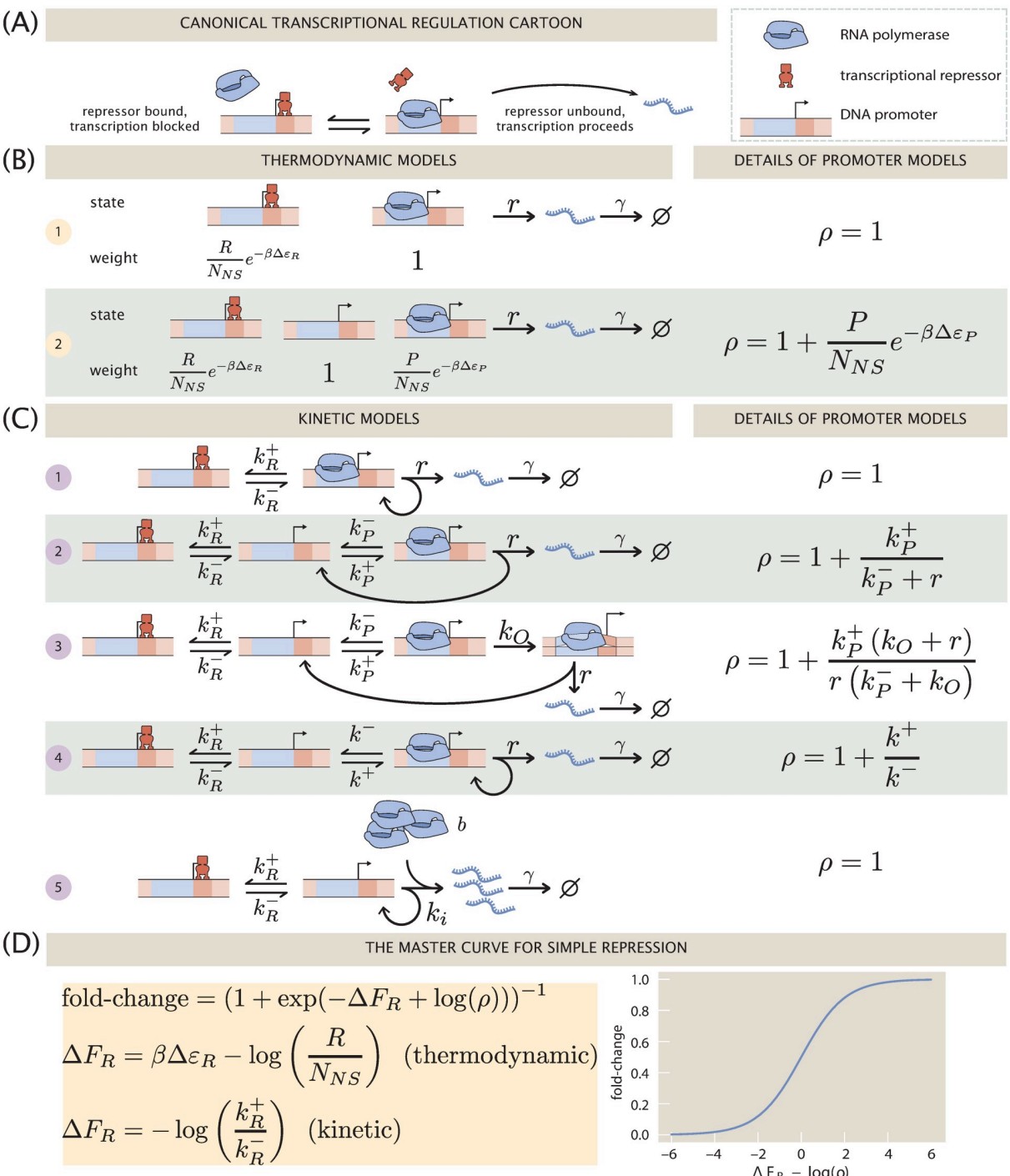

**Fig 1. An overview of the simple repression motif at the level of means.** (A) Schematic of the qualitative biological picture of the simple repression genetic architecture. (B) and (C) A variety of possible mathematicized cartoons of simple repression, along with the effective parameter $\rho$ which subsumes all regulatory details of the architecture that do not directly involve the repressor. (B) Simple repression models from a thermodynamic perspective. (C) Equivalent models cast in chemical kinetics language. (D) The "master curve" to which all cartoons in (B) and (C) collapse.

(due to effects such as supercoiling of DNA [37, 38] or DNA looping [39]) is analogous to how the Monod-Wyman-Changeux model of allostery coarse-grains continuous molecule conformations into a finite number of states [40]. The models are probabilistic with each state assigned some probability and the overall rate of transcription given by

$$\text{average rate of transcription} = \sum_i r_i p_i, \tag{1}$$

where $i$ labels the distinct states, $p_i$ is the probability of the $i^{\text{th}}$ state, and $r_i$ is the rate of transcription of that state. Ultimately, the different models differ along several key aspects: what states to consider and how to compute the probabilities of those states.

The first class of models that are the subject of the present section focus on predicting the mean level of gene expression. These models, sometimes known as thermodynamic models, invoke the tools of equilibrium statistical mechanics to compute the probabilities of the promoter microstates [5–11, 13–15]. As seen in Fig 1B, even within the class of thermodynamic models, we can make different commitments about the underlying microscopic states of the promoter. Model 1 considers only two states: a state in which a repressor (with copy number $R$) binds to an operator and a transcriptionally active state. The free energy difference between the repressor binding the operator, i.e. a specific binding site, and one of the $N_{NS}$ non-specific sites is given by $\Delta \varepsilon_R$ (given in $k_B T$ units with $\beta \equiv (k_B T)^{-1}$). Model 2 expands this model to include an empty promoter where no transcription occurs, as well as a state in which one of the $P$ RNAPs binds to the promoter with binding energy $\Delta \varepsilon_P$. Indeed, the list of options considered here does not at all exhaust the suite of different microscopic states we can assign to the promoter. The essence of thermodynamic models is to assign a discrete set of states and to use equilibrium statistical mechanics to compute the probabilities of occupancy of those states.

The second class of models that allow us to access the mean gene expression use chemical master equations to compute the probabilities of the different microscopic states [16–23]. The main differences between both modeling approaches can be summarized as: 1) Although for both classes of models the steps involving transcriptional events are assumed to be strictly irreversible, thermodynamic models force the regulation, i.e., the control over the expression exerted by the repressor, to be in equilibrium. This does not need to be the case for kinetic models. 2) Thermodynamic models ignore the mRNA count from the state of the Markov process, while kinetic models keep track of both the promoter state and the mRNA count. 3) Finally, thermodynamic and kinetic models coarse-grain to different degrees the molecular mechanisms through which RNAP enters the transcriptional event. As seen in Fig 1C, we consider a host of different kinetic models, each of which will have its own result for both the mean (this section) and noise (next section) in gene expression.

**1.1 Fold-changes are indistinguishable across models.** As a first stop on our search for the "right" model of simple repression, let us consider what we can learn from theory and experimental measurements on the average level of gene expression in a population of cells. One experimental strategy that has been particularly useful (if incomplete since it misses out on gene expression dynamics) is to measure the fold-change in mean expression [25]. The fold-change $FC$ is defined as

$$FC(R) = \frac{\langle \text{gene expression with } R > 0 \rangle}{\langle \text{gene expression with } R = 0 \rangle} = \frac{\langle m(R) \rangle}{\langle m(0) \rangle} = \frac{\langle p(R) \rangle}{\langle p(0) \rangle}, \tag{2}$$

where angle brackets $\langle \cdot \rangle$ denote the average over a population of cells and mean mRNA $\langle m \rangle$ and mean protein $\langle p \rangle$ are viewed as a function of repressor copy number $R$. What this means is that the fold-change in gene expression is a relative measurement of the effect of the

transcriptional repressor ($R > 0$) on the gene expression level compared to an unregulated promoter ($R = 0$). The third equality in Eq 2 follows from assuming that the translation efficiency, i.e., the number of proteins translated per mRNA, is the same in both conditions. In other words, we assume that mean protein level is proportional to mean mRNA level, and that the proportionality constant is the same in both conditions and therefore cancels out in the ratio. This is reasonable since the cells in the two conditions are identical except for the presence of the transcription factor, and the model assumes that the transcription factor has no direct effect on translation.

Fold-change has proven a very convenient observable in past work [41–44]. Part of its utility in dissecting transcriptional regulation is its ratiometric nature, which removes many secondary effects that are present when making an absolute gene expression measurement. Also, by measuring otherwise identical cells with and without a transcription factor present, any biological noise common to both conditions can be made to cancel out. Fig 1B and 1C depicts a smorgasbord of mathematicized cartoons for simple repression using both thermodynamic and kinetic models, respectively, that have appeared in previous literature. For each cartoon, we calculate the fold-change in mean gene expression as predicted by that model, deferring most algebraic details to the S1 Supporting Information. What we will find is that for all cartoons the fold-change can be written as a Fermi function of the form

$$FC(R) = (1 + \exp(-\Delta F_R(R) + \log(\rho)))^{-1}, \tag{3}$$

where the effective free energy contains two terms: the parameters $\Delta F_R$, an effective free energy parametrizing the repressor-DNA interaction, and $\rho$, a term derived from the level of coarse-graining used to model all repressor-free states. In other words, the effective free energy of the Fermi function can be written as the additive effect of the regulation given by the repressor via $\Delta F_R$, and the kinetic scheme used to describe the steps that lead to a transcriptional event via $\log(\rho)$ (See Fig 1D, left panel). This implies all models collapse to a single master curve as shown in Fig 1D. We will offer some intuition for why this master curve exists and discuss why at the level of the mean expression, we are unable to discriminate "right" from "wrong" cartoons given only measurements of fold-changes in expression.

*1.1.1 Two- and three-state thermodynamic models*

We begin our analysis with models 1 and 2 in Fig 1B. In each of these models the promoter is idealized as existing in a set of discrete states; the difference being whether or not the RNAP bound state is included or not. Gene expression is then assumed to be proportional to the probability of the promoter being in either the empty state (model 1) or the RNAP-bound state (model (2)). We direct the reader to the S1 Supporting Information for details on the derivation of the fold-change. For our purposes here, it suffices to state that the functional form of the fold-change for model 1 is

$$FC(R) = \left(1 + \frac{R}{N_{NS}} e^{-\beta \Delta \varepsilon_R}\right)^{-1}, \tag{4}$$

where $R$ is the number of repressors per cell, $N_{NS}$ is the number of non-specific binding sites where the repressor can bind, $\Delta \varepsilon_R$ is the repressor-operator binding energy, and $\beta \equiv (k_B T)^{-1}$. This equation matches the form of the master curve in Fig 1D with $\rho = 1$ and $\Delta F_R = \beta \Delta \varepsilon_R - \log$

$(R/N_{NS})$. For model 2 we have a similar situation. The fold-change takes the form

$$FC(R) \quad = \quad \left(1 + \frac{\frac{R}{N_{NS}}e^{-\beta\Delta\varepsilon_R}}{1 + \frac{P}{N_{NS}}e^{-\beta\Delta\varepsilon_P}}\right)^{-1} \tag{5}$$

$$= \quad (1 + \exp(-\Delta F_R + \log \rho))^{-1}, \tag{6}$$

where $P$ is the number of RNAP per cell, and $\Delta\varepsilon_P$ is the RNAP-promoter binding energy. For this model we have $\Delta F_R = \beta\Delta\varepsilon_R - \log(R/N_{NS})$ and $\rho = 1 + \frac{P}{N_{NS}}e^{-\beta\Delta\varepsilon_P}$. Thus far, we see that the two thermodynamic models, despite making different coarse-graining commitments, result in the same functional form for the fold-change in mean gene expression. We now explore how kinetic models fare when faced with computing the same observable.

### 1.1.2 Kinetic models

One of the main difference between models shown in Fig 1C, cast in the language of chemical master equations, compared with the thermodynamic models discussed in the previous section is the probability space over which they are built. Rather than keeping track only of the microstate of the promoter, and assuming that gene expression is proportional to the probability of the promoter being in a certain microstate, chemical master equation models are built on the entire probability state of both the promoter microstate, and the current mRNA count. Therefore, in order to compute the fold-change, we must compute the mean mRNA count on each of the promoter microstates, and add them all together [32].

Again, we consign all details of the derivation to the S1 Supporting Information. Here we just highlight the general findings for all five kinetic models. As already shown in Fig 1C and 1D, all the kinetic models explored can be collapsed onto the master curve. Given that the repressor-bound state only connects to the rest of the promoter dynamics via its binding and unbinding rates, $k_R^+$ and $k_R^-$ respectively, all models can effectively be separated into two categories: a single repressor-bound state, and all other promoter states with different levels of coarse graining. This structure then guarantees that, at steady-state, detailed balance between these two groups is satisfied. What this implies is that the steady-state distribution of each of the non-repressor states has the same functional form with or without the repressor, allowing us to write the fold-change as a product of the ratio of the binding and unbinding rates of the promoter, and the promoter details. This results in a fold-change of the form

$$FC \quad = \quad \left(1 + \frac{k_R^+}{k_R^-}\rho\right)^{-1}, \tag{7}$$

$$= \quad (1 + \exp(-\Delta F_R + \log(\rho)))^{-1}, \tag{8}$$

where $\Delta F_R \equiv -\log(k_R^+/k_R^-)$, and the functional forms of $\rho$ for each model change as shown in Fig 1C. Another intuitive way to think about these two terms is as follows: in all kinetic models shown in Fig 1C the repressor-bound state can only be reached from a single repressor-free state. The ratio of these two states --repressor-bound and adjacent repressor-free state-- must remain the same for all models, regardless of the details included in other promoter states if $\Delta F_R$ represents an effective free energy of the repressor binding the DNA operator. The presence of other states then draws probability density from the promoter being in either of these two states, making the ratio between the repressor-bound state and *all* repressor-free states different. The log difference in this ratio is given by $\log(\rho)$. Since model 1 and model 5 of Fig 1C consist of a single repressor-free state, $\rho$ is then necessarily 1 (See the S1 Supporting Information for further details).

The key outcome of our analysis of the models in Fig 1 is the existence of a master curve shown in Fig 1D to which the fold-change predictions of all the models collapse. This master curve is parametrized by only two effective parameters: $\Delta F_R$, which characterizes the number of repressors and their binding strength to the DNA, and $\rho$, which characterizes all other features of the promoter architecture. The key assumption underpinning this result is that no transcription occurs when a repressor is bound to its operator. Given this outcome, i.e., the degeneracy of the different models at the level of fold-change, a mean-based metric such as the fold-change that can be readily measured experimentally is insufficient to discern between these different levels of coarse-graining. The natural extension that the field has followed for the most part is to explore higher moments of the gene expression distribution in order to establish if those contain the key insights into the mechanistic nature of the gene transcription process [24, 35]. Following a similar trend, in the next section we extend the analysis of the models to higher moments of the mRNA distribution as we continue to examine the discriminatory power of these different models.

## 2 Beyond means in gene expression

In this section, our objective is to explore the same models considered in the previous section, but now with reference to the question of how well they describe the distribution of gene expression levels, with special reference to the variance in these distributions. To that end, we repeat the same pattern as in the previous section by examining the models one by one. In particular we will focus on the Fano factor, defined as the variance/mean. This metric serves as a powerful discriminatory tool to compare our different models to the null model that the steady-state mRNA distribution must be Poisson, resulting a Fano factor of one.

**2.1 Kinetic models for unregulated promoter noise.** Before we can tackle simple repression, we need an adequate phenomenological model of constitutive expression. The literature abounds with options from which we can choose, and we show several potential kinetic models for constitutive promoters in Fig 2A. Let us consider the suitability of each model for our purposes in turn.

*2.1.1 Poisson noise promoter*

The simplest model of constitutive expression that we can imagine is shown as model 1 in Fig 2A and assumes that transcripts are produced as a Poisson process from a single promoter state. This is the picture from Jones et. al. [33] that was used to interpret a systematic study of gene expression noise over a series of promoters designed to have different strengths, but no regulation. This model insists that the "true" steady-state mRNA distribution is Poisson, implying the Fano factor $\nu$ must be 1. In [33], the authors carefully attribute measured deviations from Fano = 1 to intensity variability in fluorescence measurements, gene copy number variation, and copy number fluctuations of the transcription machinery, e.g., RNAP itself. In this picture, all the corrections to Poisson behavior are derived as additive corrections to the Fano factor. This picture is appealing in its simplicity, with only two parameters, the initiation rate $r$ and degradation rate $\gamma$. In other words, the model is not excessively complex for the data at hand. But for many interesting questions, for instance in the recent work [47], attributing all deviations from the model to extrinsic noise sources, limits the kinds of predictions that can be done. To make progress then we need a (slightly) more complex model than model 1 that would allow us to incorporate the non-Poissonian features of constitutive promoters directly into a master equation formulation.

*2.1.2 Sub-Poissoninan noise promoters with RNAP escape*

A natural extension of the one-state promoter studied in the previous section is to explicitly include an empty promoter state. This state allows for single RNAP to bind and unbind from the promoter with rates $k_P^+$ and $k_P^-$, respectively, before engaging in a transcriptional event.

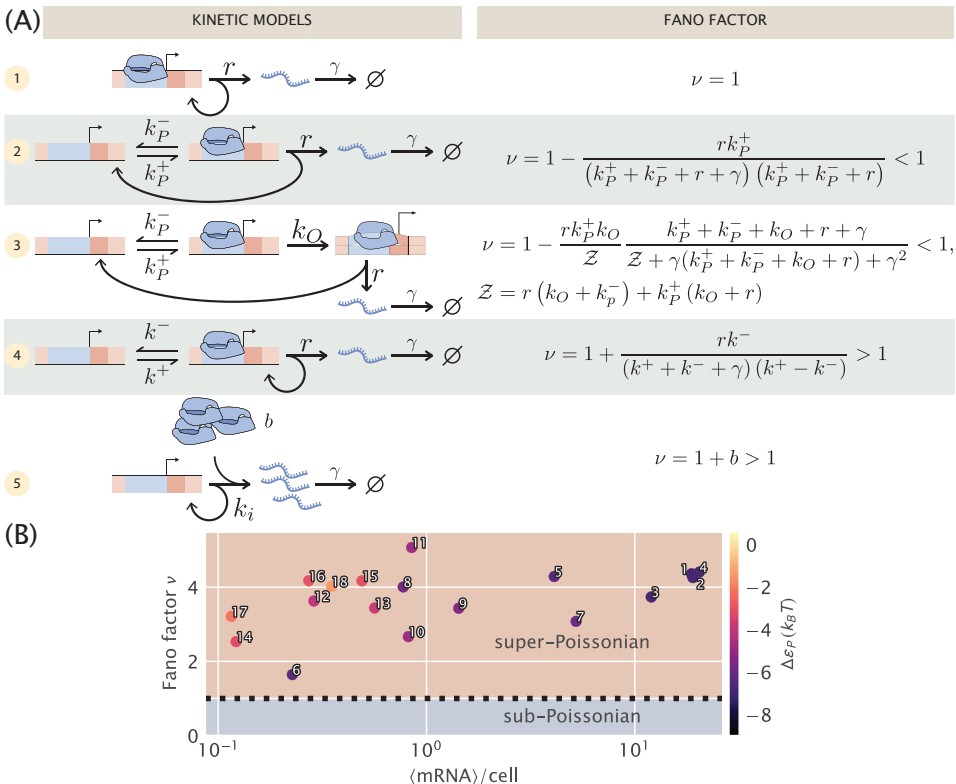

**Fig 2. Comparison of different models for noise in the constitutive promoter.** (A) The left column depicts various plausible models for the dynamics of constitutive promoters. In model (1), transcripts are produced in a Poisson process [32, 33]. Model (2) features explicit treatment of RNAP binding/unbinding kinetics [45]. Model (3) is a more detailed generalization of model (2), treating transcription initiation as a multi-step process proceeding through closed and open complexes [46]. Model (4) is somewhat analogous to (2) except with the precise nature of active and inactive states left ambiguous [17, 21, 47]. Finally, model (5) can be viewed as a certain limit of model (4) in which transcripts are produced in bursts, and initiation of bursts is a Poisson process. The right column shows the Fano factor $\nu$ (variance/mean) for each model. Note especially the crucial diagnostic: (2) and (3) have $\nu$ strictly below 1, while only for (4) and (5) can $\nu$ exceed 1. Models with Fano factors $\leq 1$ cannot produce the single-cell data observed in part (B) without introducing additional assumptions and model complexity. (B) Data from [33]. Mean mRNA count vs. Fano factor (variance/mean) for different promoters as determined with single-molecule mRNA Fluorescence *in situ* Hybridization. The colorbar indicates the predicted binding affinity of RNAP to the promoter sequence as determined in [48]. Numbers serve for cross comparison with data presented in Fig 3.

Once the RNAP irreversibly escapes from the promoter to transcribe the mRNA, the promoter goes back to this empty state. Models 2 and 3 from Fig 2A show two different versions of this process. In model 2 the polymerase can directly engage in a transcription event with rate $r$ after binding to the promoter, while model 3 includes an irreversible step with rate $k_O$ that coarse-grains the known multi-step process involved in transcriptional initiation [49].

Although physically intuitive given our understanding of how a single transcript is produced, we are forced to discard both of these models on the basis that the Fano factor for both of them is strictly $< 1$ as shown in Fig 2A (See S1 Supporting Information for detailed derivations). This is in disagreement with the experimental data from [33], reproduced in Fig 2B. All of these unregulated promoters –which only differ in their promoter sequence– show strictly super-Poissonian noise (Fano factor $> 1$). In fact, we suspect *any* model in which transcription proceeds through a multistep cycle must necessarily have $\nu < 1$. The intuitive argument compares the waiting time distribution to traverse the cycle with the waiting time for a Poisson promoter (model 1) with the same mean time. The latter is simply an exponential distribution. The former is a convolution of multiple exponentials, and intuitively the waiting time

distribution for a multistep process should be more peaked with a smaller fractional width than a single exponential with the same mean. A less disperse waiting time distribution means transcription initiations are more uniformly distributed in time relative to a Poisson process. Hence the distribution of mRNA over a population of cells should be less variable compared to Poisson, giving $v < 1$. (In the S1 Supporting Information we present a more precise version of the intuitive arguments in this paragraph).

*2.1.3 Noise in a two-state promoter with "active" and "inactive" states*

Inspired by [47], we next revisit an analog of model 2 in Fig 2A, but as with the analogous models considered in Section 1, the interpretation of the two states is changed. Rather than explicitly viewing them as RNAP bound and unbound, we view them as "active" and "inactive," which are able and unable to initiate transcripts, respectively. We are noncommittal as to the microscopic details of these states.

One interpretation [37, 38, 50] for the active and inactive states is that they represent the promoter's supercoiling state: transitions to the inactive state are caused by accumulation of positive supercoiling, which inhibits transcription, and transitions back to "active" are caused by gyrase or other topoisomerases relieving the supercoiling. This is an interesting possibility because it would mean the timescale for promoter state transitions is driven by topoisomerase kinetics, not by RNAP kinetics. From in vitro measurements, the former are suggested to be of order minutes [37]. Contrast this with model 2, where the state transitions are assumed to be governed by RNAP, which, assuming a copy number per cell of order $10^3$, has a diffusion-limited association rate $k_{on} \sim 10^2$ s$^{-1}$ to a target promoter. Combined with known $K_d$'s of order $\mu$M, this gives an RNAP dissociation rate $k_{off}$ of order $10^2$ s$^{-1}$. As we will show below, however, there are some lingering puzzles with interpreting this supercoiling hypothesis, so we leave it as a speculative hypothesis and refrain from assigning definite physical meaning to the two states in this model.

Intuitively one might expect that, since transcripts are produced as a Poisson process only when the promoter is in one of the two states in this model, transcription initiations should now be "bunched" in time, in contrast to the "anti-bunching" of models 2 and 3 above. One might further guess that this bunching would lead to super-Poissonian noise in the mRNA distribution over a population of cells. Indeed, as shown in the S1 Supporting Information, a calculation of the Fano factor produces

$$v = 1 + \frac{rk^-}{(k^+ + k^- + \gamma)(k^+ + k^-)}, \tag{9}$$

which is strictly greater than 1, verifying the above intuition. Note we have dropped the *P* label on the promoter switching rates to emphasize that these very likely do not represent kinetics of RNAP itself. This calculation can also be sidestepped by noting that the model is mathematically equivalent to the simple repression model from [33], with states and rates relabeled and reinterpreted.

How does this model compare to model 1 above? In model 1, all non-Poisson features of the mRNA distribution were handled as extrinsic corrections. By contrast, here the 3 parameter model is used to fit the full mRNA distribution as measured in mRNA FISH experiments. In essence, all variability in the mRNA distribution is regarded as "intrinsic," arising either from stochastic initiation or from switching between the two coarse-grained promoter states. The advantage of this approach is that it fits neatly into the master equation picture, and the parameters thus inferred can be used as input for more complicated models with regulation by transcription factors.

While this seems promising, there is a major drawback for our purposes which was already uncovered by the authors of [47]: the statistical inference problem is nonidentifiable, in the

sense described in Section 4.3 of [51]. What this means is that it is impossible to infer the parameters $r$ and $k^-$ from the single-cell mRNA counts data of [33] (as shown in S2 Fig of [47]). Rather, only the ratio $r/k^-$ could be inferred. In that work, the problem was worked around with an informative prior on the ratio $k^-/k^+$. That approach is unlikely to work here, as, recall, our entire goal in modeling constitutive expression is to use it as the basis for a yet more complicated model, when we add on repression. But adding more complexity to a model that is already poorly identified is a fool's errand, so we will explore one more potential model.

*2.1.4 Noise model for one-state promoter with explicit bursts*

The final model we consider is inspired by the failure mode of model 4. The key observation above was that, as found in [47], only two parameters, $k^+$ and the ratio $r/k^-$, could be directly inferred from the single-cell mRNA data from [33]. So let us take this seriously and imagine a model where these are the only two model parameters. What would this model look like?

To develop some intuition, consider model 4 in the limit $k^+ \ll k^- \lesssim r$, which is roughly satisfied by the parameters inferred in [47]. In this limit, the system spends the majority of its time in the inactive state, occasionally becoming active and making a burst of transcripts. This should call to mind the well-known phenomenon of transcriptional bursting, as observed in, e.g., [37, 38, 52–54]. Let us make this correspondence more precise. The mean dwell time in the active state is $1/k^-$. While in this state, transcripts are produced at a rate $r$ per unit time. So on average, $r/k^-$ transcripts are produced before the system switches to the inactive state. Once in the inactive state, the system dwells there for an average time $1/k^+$ before returning to the active state and repeating the process. $r/k^-$ resembles an average burst size, and $1/k^+$ resembles the time interval between burst events. More precisely, $1/k^+$ is the mean time between the end of one burst and the start of the next, whereas $1/k^+ + 1/k^-$ would be the mean interval between the start of two successive burst events, but in the limit $k^+ \ll k^-$, $1/k^+ + 1/k^- \approx 1/k^+$. Note that this limit ensures that the waiting time between bursts is approximately exponentially distributed, with mean set by the only timescale left in the problem, $1/k^+$. If instead it were the case that $k^+ \sim k^-$, then the waiting time $1/k^+ + 1/k^-$ would have a peaked distribution, but this does not appear to be the case for any of datasets from [33].

Let us now verify this intuition with a precise derivation to check that $r/k^-$ is in fact the mean burst size and to obtain the full burst size distribution. Consider first a constant, known dwell time $T$ in the active state. Transcripts are produced at a rate $r$ per unit time, so the number of transcripts $n$ produced during $T$ is Poisson distributed with mean $rT$, i.e.,

$$P(n \mid T) = \frac{(rT)^n}{n!} \exp(-rT). \tag{10}$$

Since the dwell time $T$ is unobservable from single molecule mRNA counts, we actually want $P(n)$, the dwell time distribution with no conditioning on $T$. Basic rules of probability theory tell us we can write $P(n)$ in terms of $P(n \mid T)$ as

$$P(n) = \int_0^\infty P(n \mid T) P(T) dT. \tag{11}$$

But we know the dwell time distribution $P(T)$, which is exponentially distributed according to

$$P(T) = k^- \exp(-Tk^-), \tag{12}$$

so $P(n)$ can be written as

$$P(n) = k^- \frac{r^n}{n!} \int_0^\infty T^n \exp[-(r + k^-)T] \, dT. \tag{13}$$

A standard integral table shows $\int_0^\infty x^n e^{-ax}\,dx = n!/a^{n+1}$, so

$$P(n) = \frac{k^- r^n}{(k^- + r)^{n+1}} = \frac{k^-}{k^- + r}\left(\frac{r}{k^- + r}\right)^n = \frac{k^-}{k^- + r}\left(1 - \frac{k^-}{k^- + r}\right)^n, \tag{14}$$

which is exactly the geometric distribution with standard parameter $\theta \equiv k^-/(k^- + r)$ and domain $n \in \{0, 1, 2, \ldots\}$. The mean of the geometric distribution, with this convention, is

$$\langle n \rangle = \frac{1 - \theta}{\theta} = \left(1 - \frac{k^-}{k^- + r}\right)\frac{k^- + r}{k^-} = \frac{r}{k^-}, \tag{15}$$

exactly as we guessed intuitively above.

So in taking the limit $r, k^- \to \infty$, $r/k^- \equiv b$, we obtain a model which effectively has only a single promoter state, which initiates bursts at rate $k^+$ (transitions to the active state, in the model 4 picture). The master equation for mRNA copy number $m$ as derived in the S1 Supporting Information takes the form

$$\frac{d}{dt}p(m, t) = \quad (m + 1)\gamma p(m + 1, t) - m\gamma p(m, t)$$
$$+ \sum_{m'=0}^{m-1} k_i p(m', t)\mathrm{Geom}(m - m'; b) - \sum_{m'=m+1}^{\infty} k_i p(m, t)\mathrm{Geom}(m' - m; b), \tag{16}$$

where we use $k_i$ to denote the burst initiation rate, $\mathrm{Geom}(n;b)$ is the geometric distribution with mean $b$, i.e., $\mathrm{Geom}(n; b) = \frac{1}{1+b}\left(\frac{b}{1+b}\right)^n$ (with domain over nonnegative integers as above). The first two terms are the usual mRNA degradation terms. The third term enumerates all ways the system can produce a burst of transcripts and arrive at copy number $m$, given that it had copy number $m'$ before the burst. The fourth term allows the system to start with copy number $m$, then produce a burst and end with copy number $m'$. In fact this last sum has trivial $m'$ dependence and simply enforces normalization of the geometric distribution. Carrying it out we have

$$\frac{d}{dt}p(m, t) = \quad (m + 1)\gamma p(m + 1, t) - m\gamma p(m, t)$$
$$+ \sum_{m'=0}^{m-1} k_i p(m', t)\mathrm{Geom}(m - m'; b) - k_i p(m, t), \tag{17}$$

We direct readers again to the S1 Supporting Information for further details. This improves on model 4 in that now the parameters are easily inferred, as we will see later, and have clean interpretations. The non-Poissonian features are attributed to the empirically well-established phenomenological picture of bursty transcription.

The big approximation in going from model 4 to 5 is that a burst is produced instantaneously rather than over a finite time. If the true burst duration is not short compared to transcription factor kinetic timescales, this could be a problem in that mean burst size in the presence and absence of repressors could change, rendering parameter inferences from the constitutive case inappropriate. Let us make some simple estimates of this.

Consider the time delay between the first and final RNAPs in a burst initiating transcription (*not* the time to complete transcripts, which potentially could be much longer.) If this timescale is short compared to the typical search timescale of transcription factors, then all is well. The estimates from deHaseth et. al. [49] put RNAP's diffusion-limited on rate around $\sim$ few $\times\, 10^{-2}\ \mathrm{nM}^{-1}\ \mathrm{s}^{-1}$ and polymerase loading as high as $1\ \mathrm{s}^{-1}$. Then for reasonable burst sizes of $< 10$, it is reasonable to guess that bursts might finish initiating on a timescale of

tens of seconds or less (with another 30-60 sec to finish elongation, but that does not matter here). A transcription factor with typical copy number of order 10 (or less) would have a diffusion-limited association rate of order $(10 \text{ sec})^{-1}$ [55]. Higher copy number TFs tend to have many binding sites over the genome, which should serve to pull them out of circulation and keep their effective association rates from rising too large [56]. Therefore, there is *perhaps* a timescale separation possible between transcription factor association rates and burst durations, but this assumption could very well break down, so we will have to keep it in mind when we infer repressor rates from the Jones et. al. single-cell mRNA counts data later [33].

In reflecting on these 5 models, the reader may feel that exploring a multitude of potential models just to return to a very minimal phenomenological model of bursty transcription may seem highly pedantic. But the purpose of the exercise was to examine a host of models from the literature and understand why they are insufficient, one way or another, for our purposes. Along the way we have learned that the detailed kinetics of RNAP binding and initiating transcription are probably irrelevant for setting the population distribution of mRNA. The timescales are simply too fast, and as we will see later in Figs 3 and 4, the noise seems to be governed by slower timescales. Perhaps in hindsight this is not surprising: intuitively, the degradation rate $\gamma$ sets the fundamental timescale for mRNA dynamics, and any other processes that substantially modulate the mRNA distribution should not differ from $\gamma$ by orders of magnitude.

## 3 Finding the "right" model: Bayesian parameter inference

In this section of the paper, we continue our program of providing one complete description of the entire broad sweep of studies that have been made in the context of the repressor-operator model, dating all the way back to the original work of Jacob and Monod and including the visionary quantitative work of Müller-Hill and collaborators [29] and up to more recent studies [41]. In addition, the aim is to reconcile the thermodynamic and kinetic perspectives that have been brought to bear on this problem. From Section 1, this reconciliation depends on a key quantitative question as codified by Eq 3: does the free energy of repressor binding, as described in the thermodynamic models and indirectly inferred from gene expression measurements, agree with the corresponding values of repressor binding and unbinding rates in the kinetic picture, measured or inferred more directly? In this section we tackle the statistical inference problem of inferring these repressor rates from single-cell mRNA counts data. But before we can turn to the full case of simple repression, we must choose an appropriate model of the constitutive promoter and infer the parameter values in that model. This is the problem we address first.

**3.1 Parameter inference for constitutive promoters.** From consideration of Fano factors in the previous section, we suspect that model 5 in Fig 2A, a one-state bursty model of constitutive promoters, achieves the right balance of complexity and simplicity, by allowing both Fano factor $\nu > 1$, but also by remedying, by design, the problems of parameter degeneracy that model 4 in Fig 2A suffered [47]. Does this stand up to closer scrutiny, namely, comparison to full mRNA distributions rather than simply their moments? We will test this thoroughly on single-cell mRNA counts for different unregulated promoters from Jones et. al. [33].

It will be instructive, however, to first consider the Poisson promoter, model 1 in Fig 2. As we alluded to earlier, since the Poisson distribution has a Fano factor $\nu$ strictly equal to 1, and all of the observed data in Fig 2B has Fano factor $\nu > 1$, we might already suspect that this model is incapable of fitting the data. We will verify that this is in fact the case. Using the same argument we can immediately rule out models 2 and 3 from Fig 2A. These models have Fano factors $\nu \leq 1$ meaning they are underdispersed relative to the Poisson distribution. We will also not explicitly consider model 4 from Fig 2A since it was already thoroughly analyzed in [47], and since model 5 can be viewed as a special case of it.

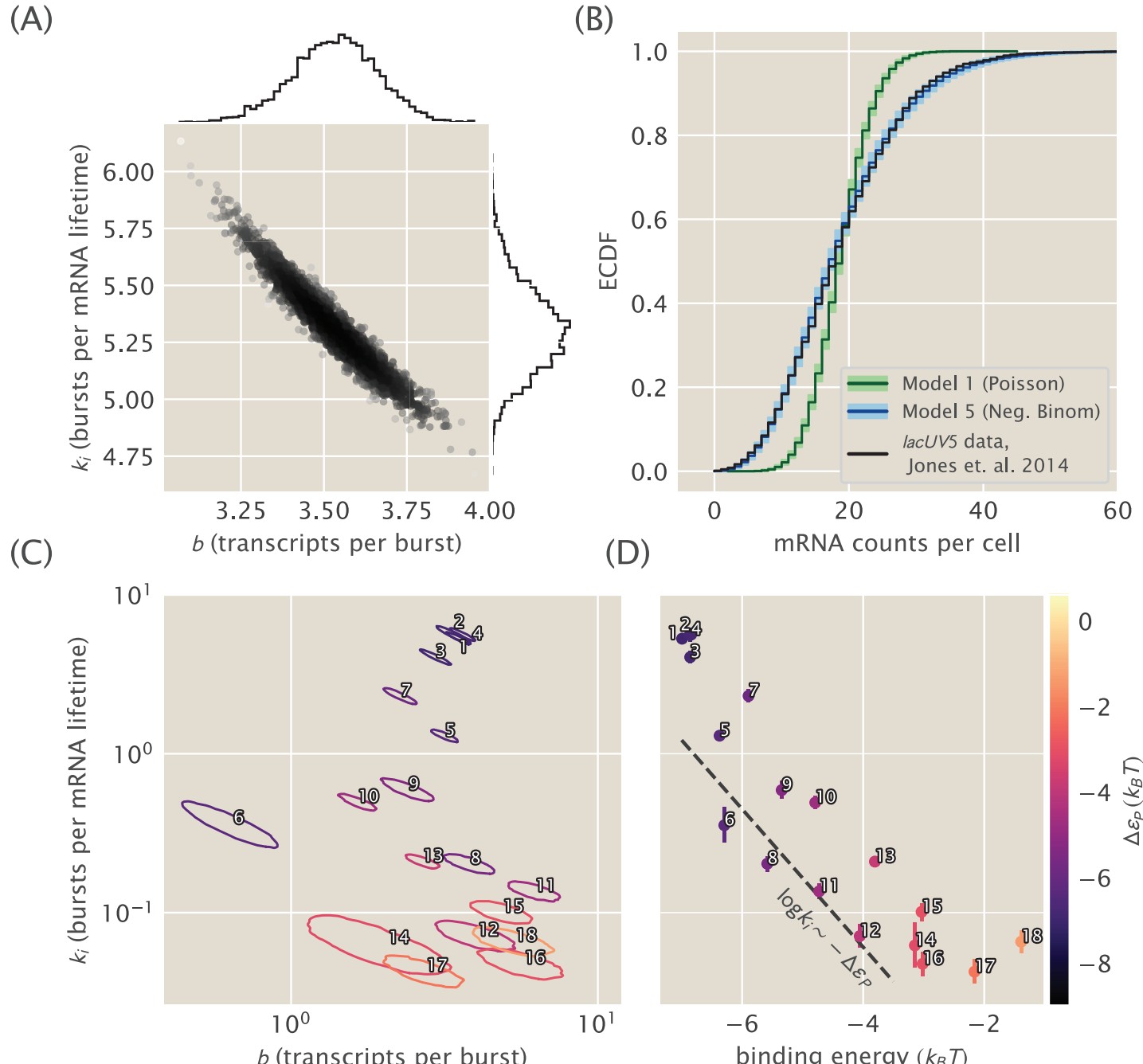

**Fig 3. Constitutive promoter posterior inference and model comparison.** (A) The joint posterior density of model 5, the bursty promoter with negative binomially-distributed steady state, is plotted with MCMC samples. 1D marginal probability densities are plotted as flanking histograms. The model was fit on *lacUV5* data from [33]. (B) The empirical cumulative distribution function (ECDF) of the observed population distribution of mRNA transcripts under the control of a constitutive *lacUV5* promoter is shown in black. The median posterior predictive ECDFs for models (1), Poisson, and (5), negative binomial, are plotted in dark green and dark blue, respectively. Lighter green and blue regions enclose 95% of all posterior predictive samples from their respective models. Model (1) is in obvious contradiction with the data while model (5) is not. Single-cell mRNA count data is again from [33]. (C) Joint posterior distributions for burst rate $k_i$ and mean burst size $b$ for 18 unregulated promoters from [33]. Each contour indicates the 95% highest posterior probability density region for a particular promoter. Note that the vertical axis is shared with (D). (D) Plots of the burst rate $k_i$ vs. the binding energy for each promoter as predicted in [48]. The dotted line shows the predicted slope according to Eq 22, described in text. Each individual promoter is labeled with a unique number in both (C) and (D) for cross comparison and for comparison with Fig 2B.

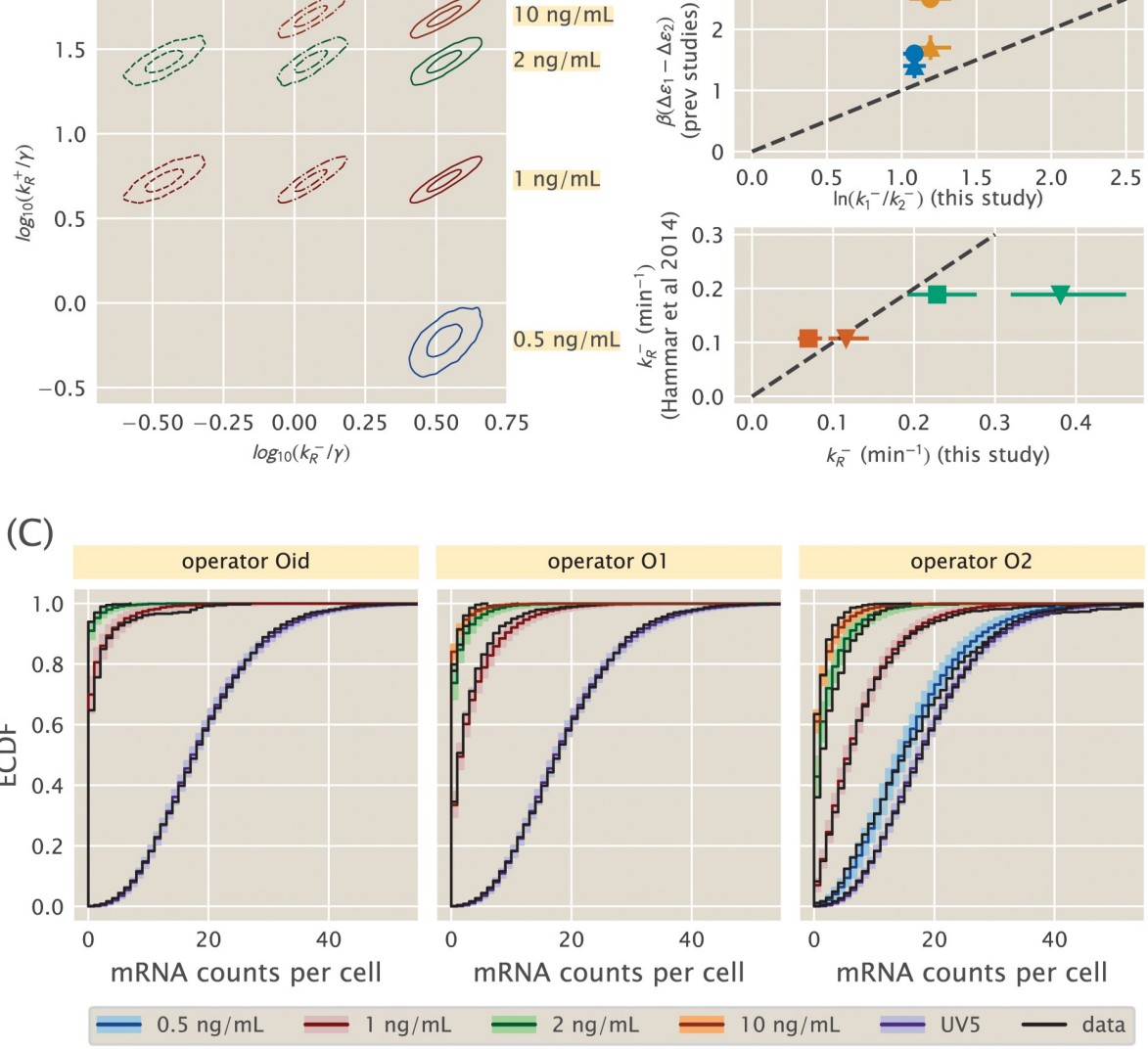

**Fig 4. Simple repression parameter inference and comparison.** (A) Contours which enclose 50% and 95% of the posterior probability mass are shown for each of several 2D slices of the 9D posterior distribution. The model assumes one unbinding rate for each operator (Oid, O1, O2) and one binding rate for each aTc induction concentration (corresponding to an unknown mean repressor copy number). (B, upper) Ratios of our inferred unbinding rates are compared with operator binding energy differences measured by Garcia and Phillips [41] (triangles) and Razo-Mejia et. al. [43] (circles). Blue glyphs compare O2-O1, while orange compare O1-Oid. Points with perfect agreement would lie on the dotted line. (B, lower) Unbinding rates for O1 (green) and Oid (red) inferred in this work are compared with single-molecule measurements from Hammar et. al. [55]. We plot the comparison assuming illustrative mRNA lifetimes of $\gamma^{-1}$ = 3 min (triangles) or $\gamma^{-1}$ = 5 min (squares). Dotted line is as in upper panel. (C) Theory-experiment comparison are shown for each of the datasets used in the inference of the model in (A). Observed single-molecule mRNA counts data from [33] are plotted as black lines. The median of the randomly generated samples for each condition is plotted as a dark colored line. Lighter colored bands enclose 95% of all samples for a given operator/repressor copy number pair. The unregulated promoter, *lacUV5*, is shown with each as a reference.

Our objective for this section will then be to assess whether or not model 5 is quantitatively able to reproduce experimental data. In other words, if our claim is that the level of coarse graining in this model is capable of capturing the relevant features of the data, then we should be able to find values for the model parameters that can match theoretical predictions with single-molecule mRNA count distributions. A natural language for this parameter inference

problem is that of Bayesian probability. We will then build a Bayesian inference pipeline to fit the model parameters to data. We will use the full dataset of single-cell mRNA counts from [33] used in Fig 2B. To gain intuition on how this analysis is done we analyzed the "wrong" model 1 in Fig 2A. For this analysis, we direct the reader to the S1 Supporting Information as an intuitive introduction to the Bayesian framework with a simpler model.

*3.1.1 Model 5—bursty promoter*

Let us now consider the problem of parameter inference for model five. As derived in the S1 Supporting Information, the steady-state mRNA distribution in this model is a negative binomial distribution [36], given by

$$p(m) = \frac{\Gamma(m + k_i)}{\Gamma(m+1)\Gamma(k_i)} \left(\frac{1}{1+b}\right)^{k_i} \left(\frac{b}{1+b}\right)^m,$$  (18)

where $b$ is the mean burst size and $k_i$ is the burst rate in units of the mRNA degradation rate $\gamma$. As sketched earlier, to think of the negative binomial distribution in terms of an intuitive "story," in the precise meaning of [57], we imagine the arrival of bursts as a Poisson process with rate $k_i$, where each burst has a geometrically-distributed size with mean size $b$.

As for the Poisson promoter model, this expression for the steady-state mRNA distribution is exactly the likelihood we want to use when stating Bayes theorem. Denoting the single-cell mRNA count data as $D = \{m_1, m_2, \ldots, m_N\}$, Bayes' theorem takes the form

$$p(k_i, b \mid D) \propto p(D \mid k_i, b)p(k_i, b).$$  (19)

We already have our likelihood --the product of $N$ negative binomials as Eq 18—so we only need to choose priors on $k_i$ and $b$. For the datasets from [33] that we are analyzing, as for the Poisson promoter model above, we are still data-rich so the prior's influence remains weak, but not nearly as weak because the dimensionality of our model has increased from one parameter to two. Details on the arguments behind our prior distribution selection are left for the S1 Supporting Information. We state here that the natural scale to explore these parameters is logarithmic. This is commonly the case for parameters for which our previous knowledge based on our domain expertise spans several orders of magnitude. For this we chose log-normal distributions for both $k_i$ and $b$. Details on the mean and variance of these distributions can be found in the S1 Supporting Information.

We carried out Markov-Chain Monte Carlo (MCMC) sampling on the posterior of this model, starting with the constitutive *lacUV5* dataset from [33]. The resulting MCMC samples are shown in Fig 3A. In contrast to the active/inactive constitutive model considered in [47] (kinetic model 4 in Fig 2A), this model is well-identified with both parameters determined to a fractional uncertainty of 5-10%. The strong correlation reflects the fact that their product sets the mean of the mRNA distribution, which is tightly constrained by the data, but there is weak "sloppiness" [58] along a set of values with a similar product.

Having found the model's posterior to be well-identified as with the Poisson promoter (S1 Supporting Information), the next step is to compare both models with experimental data. To do this for the case of the bursty promoter, for each of the parameter samples shown in Fig 3A we generated negative bionomial-distributed mRNA counts. As MCMC samples parameter space proportionally to the posterior distribution, this set of random samples span the range of possible values that we would expect given the correspondence between our theoretical model and the experimental data. A similar procedure can be applied to the Poisson promoter. To compare so many samples with the actual observed data, we can use empirical cumulative distribution functions (ECDF) of the distribution quantiles. This representation is shown in Fig 3B. In this example, the median for each possible mRNA count for the Poisson distribution is

shown as a dark green line, while the lighter green contains 95% of the randomly generated samples. This way of representing the fit of the model to the data gives us a sense of the range of data we might consider plausible, under the assumption that the model is true. For this case, as we expected given our premise of the Poisson promoter being wrong, it is quite obvious that the observed data, plotted in black is not consistent with the Poisson promoter model. An equivalent plot for the bursty promoter model is shown in blue. Again the darker tone shows the median, while the lighter color encompasses 95% of the randomly generated samples. Unlike the Poisson promoter model, the experimental ECDF closely tracks the posterior predictive ECDF, indicating this model is actually able to generate the observed data and increasing our confidence that this model is sufficient to parametrize the physical reality of the system.

The commonly used promoter sequence *lacUV5* is our primary target here, since it forms the core of all the simple repression constructs of [33] that we consider in Section 3.2. Nevertheless, we thought it wise to apply our bursty promoter model to the other 17 unregulated promoters available in the single-cell mRNA count dataset from [33] as a test that the model is capturing the essential phenomenology. If the model fit well to all the different promoters, this would increase our confidence that it would serve well as a foundation for inferring repressor kinetics later in Section 3.2. Conversely, were the model to fail on more than a couple of the other promoters, it would give us pause.

Fig 3C shows the results, plotting the posterior distribution from individually MCMC sampling all 18 constitutive promoter datasets from [33]. To aid visualization, rather than plotting samples for each promoter's posterior as in Fig 3A, for each posterior we find and plot the curve that surrounds the 95% highest probability density region. What this means is that each contour encloses approximately 95% of the samples, and thus 95% of the probability mass, of its posterior distribution. Theory-experiment comparisons, shown in the S1 Supporting Information, display a similar level of agreement between data and predictive samples as for the bursty model with *lacUV5* in Fig 3B.

One interesting feature from Fig 3C is that burst rate varies far more widely, over a range of $\sim 10^2$, than burst size, confined to a range of $\lesssim 10^1$ (and with the exception of promoter 6, just a span of 3 to 5-fold). This suggests that $k_i$, not $b$, is the key dynamic variable that promoter sequence tunes.

*3.1.2 Connecting inferred parameters to prior work*

It is interesting to connect these inferences on $k_i$ and $b$ to the work of [48], where these same 18 promoters were considered through the lens of the three-state thermodynamic model (model 2 in Fig 1B) and binding energies $\Delta\varepsilon_P$ were predicted from an energy matrix model derived from [59]. As previously discussed the thermodynamic models of gene regulation can only make statements about the mean gene expression. This implies that we can draw the connection between both frameworks by equating the mean mRNA $\langle m \rangle$. This results in

$$\langle m \rangle = \frac{k_i b}{\gamma} = \frac{r}{\gamma} \frac{\frac{P}{N_{NS}} \exp(-\beta\Delta\varepsilon_P)}{1 + \frac{P}{N_{NS}} \exp(-\beta\Delta\varepsilon_P)}. \tag{20}$$

By taking the weak promoter approximation for the thermodynamic model ($P/N_{NS} \exp(-\beta\Delta\varepsilon_P) \ll 1$) results in [48]

$$\langle m \rangle = \frac{k_i b}{\gamma} = \frac{r}{\gamma} \frac{P}{N_{NS}} \exp(-\beta\Delta\varepsilon_P), \tag{21}$$

valid for all the binding energies considered here.

Given this result, how are the two coarse-grainings related? A quick estimate can shed some light. Consider for instance the *lacUV5* promoter, which we see from Fig 3A has $k_i/\gamma \sim b \sim$ few, from Fig 3B has $\langle m \rangle \sim 20$, and from [48] has $\beta\Delta\varepsilon_P \sim -6.5$. Further we generally assume $P/N_{NS} \sim 10^{-3}$ since $N_{NS} \approx 4.6 \times 10^6$ and $P \sim 10^3$. After some guess-and-check with these values, one finds the only association that makes dimensional sense and produces the correct order-of-magnitude for the known parameters is to take

$$\frac{k_i}{\gamma} = \frac{P}{N_{NS}} \exp(-\beta\Delta\varepsilon_P) \qquad (22)$$

and

$$b = \frac{r}{\gamma}. \qquad (23)$$

Fig 3D shows that this linear scaling between $\ln k_i$ and $-\beta\Delta\varepsilon_P$ is approximately true for all 18 constitutive promoters considered. The plotted line is simply Eq 22 and assumes $P \approx 5000$.

While the associations represented by Eqs 22 and 23 appear to be borne out by the data in Fig 3, we do not find the association of parameters they imply to be intuitive. We are also cautious to ascribe too much physical reality to the parameters. Indeed, part of our point in comparing the various constitutive promoter models is to demonstrate that these models each provide an internally self-consistent framework that adequately describes the data, but attempting to translate between models reveals the dubious physical interpretation of their parameters.

Our remaining task in this work is a determination of the physical reality of thermodynamic binding energies in Fig 1, as codified by the thermodynamic-kinetic equivalence of Eq 3. For our phenomenological needs here model 5 in Fig 2A is more than adequate: the posterior distributions in Fig 3C are cleanly identifiable and the predictive checks in the S1 Supporting Information indicate no discrepancies between the model and the mRNA single-molecule count data of [33]. Of the models we have considered it is unique in satisfying both these requirements. As a result we will use it as a foundation to build upon in the next section when we add regulation.

**3.2 Transcription factor kinetics can be inferred from single-cell mRNA distribution measurements.** *3.2.1 Building the model and performing parameter inference*

Now that we have a satisfactory model in hand for constitutive promoters, we would like to return to the main thread: can we reconcile the thermodynamic and kinetic models by putting to the test Eq 3, the correspondence between indirectly inferred thermodynamic binding energies and kinetic rates? To make this comparison, is it possible to infer repressor binding and unbinding rates from mRNA distributions over a population of cells as measured by single-molecule Fluorescence *in situ* Hybridization in [33]? If so, how do these inferred rates compare to direct single-molecule measurements such as from [55] and to binding energies such as from [41] and [43], which were inferred under the assumptions of the thermodynamic models in Fig 1B? And can this comparison shed light on the unreasonable effectiveness of the thermodynamic models, for instance, in their application in [44, 60]?

As we found in Section 2, for our purposes the "right" model of a constitutive promoter is the bursty picture, model five in Fig 2A. Therefore our starting point here is the analogous model with repressor added, model 5 in Fig 1C. For a given repressor binding site and copy number, this model has four rate parameters to be inferred: the repressor binding and unbinding rates $k_R^+$, and $k_R^-$, the initiation rate of bursts, $k_i$, and the mean burst size $b$ (we nondimensionalize all of these by the mRNA degradation rate $\gamma$).

Before showing the mathematical formulation of our statistical inference model, we would like to sketch the intuitive structure. The dataset from [33] we consider consists of single-cell mRNA counts data of nine different conditions, spanning several combinations of three unique repressor binding sites (the so-called Oid, O1, and O2 operators) and four unique repressor copy numbers. We assume that the values of $k_i$ and $b$ are known, since we have already cleanly inferred them from constitutive promoter data, and further we assume that these values are the same across datasets with different repressor binding sites and copy numbers. In other words, we assume that the regulation of the transcription factor does not affect the mean burst size nor the burst initiation rate. The regulation occurs as the promoter is taken away from the transcriptionally active state when the promoter is bound by repressor. We assume that there is one unbinding rate parameter for each repressor binding site, and likewise one binding rate for each unique repressor copy number. This makes our model seven dimensional, or nine if one counts $k_i$ and $b$ as well. Note that we use only a subset of the datasets from Jones et. al. [33], as discussed more in the S1 Supporting Information.

Formally now, denote the set of seven repressor rates to be inferred as

$$\vec{k} = \{k_{Oid}^-, k_{O1}^-, k_{O2}^-, k_{0.5}^+, k_1^+, k_2^+, k_{10}^+\},\tag{24}$$

where subscripts for dissociation rates $k^-$ indicate the different repressor binding sites, and subscripts for association rates $k^+$ indicate the concentration of the small-molecule that controlled the expression of the LacI repressor (see the S1 Supporting Information). This is because for this particular dataset the repressor copy numbers were not measured directly, but it is safe to assume that a given concentration of the inducer resulted in a specific mean repressor copy number [60]. Also note that the authors of [33] report estimates of LacI copy number per cell rather than direct measurements. However, these estimates were made assuming the validity of the thermodynamic models in Fig 1, and since testing these models is our present goal, it would be circular logic if we were to make the same assumption. Therefore we will make no assumptions about the LacI copy number for a given inducer concentration.

Having stated the problem, Bayes' theorem reads

$$p(\vec{k}, k_i, b \mid D) \propto p(D \mid \vec{k}, k_i, b)p(\vec{k}, k_i, b),\tag{25}$$

where $D$ is again the set of all $N$ observed single-cell mRNA counts across the various conditions. We assume that individual single-cell measurements are independent so that the likelihood factorizes as

$$p(D \mid \vec{k}, k_i, b) = \prod_{j=1}^{N} p(m \mid \vec{k}, k_i, b) = \prod_{j=1}^{N} p(m \mid k_j^+, k_j^-, k_i, b)\tag{26}$$

where $k_j^\pm$ represent the appropriate binding and unbinding rates out of $\vec{k}$ for the $j$-th measured cell. The probability $p(m \mid k_j^+, k_j^-, k_i, b)$ appearing in the last expression is exactly Eq. S205, the steady-state distribution for our bursty model with repression derived in the S1 Supporting Information, which for completeness we reproduce here as

$$p(m \mid k_R^+, k_R^-, k_i, b) = \frac{\Gamma(\alpha + m)\Gamma(\beta + m)\Gamma(k_R^+ + k_R^-)}{\Gamma(\alpha)\Gamma(\beta)\Gamma(k_R^+ + k_R^- + m)} \frac{b^m}{m!}$$
$$\times {}_2F_1(\alpha + m, \beta + m, k_R^+ + k_R^- + m; -b).\tag{27}$$

where ${}_2F_1$ is the confluent hypergeometric function of the second kind and $\alpha$ and $\beta$, defined

for notational convenience, are

$$\alpha = \frac{1}{2}\left(k_i + k_R^- + k_R^+ + \sqrt{(k_i + k_R^- + k_R^+)^2 - 4k_i k_R^-}\right)$$

$$\beta = \frac{1}{2}\left(k_i + k_R^- + k_R^+ - \sqrt{(k_i + k_R^- + k_R^+)^2 - 4k_i k_R^-}\right). \tag{28}$$

This likelihood is rather inscrutable. We did not find any of the known analytical approximations for $_2F_1$ useful in gaining intuition, so we instead resorted to numerics. One insight we found was that for very strong or very weak repression, the distribution in Eq 27 is well approximated by a negative binomial with burst size $b$ and burst rate $k_i$ equal to their constitutive *lacUV5* values, except with $k_i$ multiplied by the fold-change $(1 + k_R^+/k_R^-)^{-1}$. In other words, once again only the ratio $k_R^+/k_R^-$ was detectable. But for intermediate repression, the distribution was visibly broadened with Fano factor greater than $1 + b$, the value for the corresponding constitutive case. This indicates that the repressor rates had left an imprint on the distribution, and perhaps intuitively, this intermediate regime occurs for values of $k_R^\pm$ comparable to the burst rate $k_i$. Put another way, if the repressor rates are much faster or much slower than $k_i$, then there is a timescale separation and effectively only one timescale remains, $k_i(1 + k_R^+/k_R^-)^{-1}$. Only when all three rates in the problem are comparable does the mRNA distribution retain detectable information about them.

Next we specify priors. As for the constitutive model, weakly informative log-normal priors are a natural choice for all our rates. We found that if the priors were too weak, our MCMC sampler would often become stuck in regions of parameter space with very low probability density, unable to move. We struck a balance in choosing our prior widths between helping the sampler run while simultaneously verifying that the marginal posteriors for each parameter were not artificially constrained or distorted by the presence of the prior. All details for our prior distributions are listed in the S1 Supporting Information.

We ran MCMC sampling on the full nine dimensional posterior specified by this model. To attempt to visualize this object, in Fig 4A we plot several two-dimensional slices as contour plots, analogous to Fig 3C. Each of these nine slices corresponds to the $(k_R^+, k_R^-)$ pair of rates for one of the conditions from the dataset used to fit the model and gives a sense of the uncertainty and correlations in the posterior. We note that the 95% uncertainties of all the rates span about $\sim 0.3$ log units, or about a factor of two, with the exception of $k_{0.5}^+$, the association rate for the lowest repressor copy number which is somewhat larger.

*3.2.2 Comparison with prior measurements of repressor binding energies*

Our primary goal in this work is to reconcile the kinetic and thermodynamic pictures of simple repression. Towards this end we would like to compare the repressor kinetic rates we have inferred with the repressor binding energies inferred through multiple methods in [41] and [43]. If the agreement is close, then it suggests that the thermodynamic models are not wrong and the repressor binding energies they contain correspond to physically real free energies, not mere fit parameters.

Fig 4B shows both comparisons, with the top panel comparing to thermodynamic binding energies and the bottom panel comparing to single-molecule measurements. First consider the top panel and its comparison between repressor kinetic rates and binding energies. As described in section 1, if the equilibrium binding energies from [41] and [43] indeed are the physically real binding energies we believe them to be, then they should be related to the repressor kinetic rates via Eq 3, which we restate here,

$$\Delta F_R = \beta \Delta \varepsilon_R - \log(R/N_{NS}) = -\log(k_R^+/k_R^-). \tag{29}$$

Assuming mass action kinetics implies that $k_R^+$ is proportional to repressor copy number $R$, or more precisely, it can be thought of as repressor copy number times some intrinsic per molecule association rate. But since $R$ is not directly known for our data from [33], we cannot use this equation directly. Instead we can consider two different repressor binding sites and compute the *difference* in binding energy between them, since this difference depends only on the unbinding rates and not on the binding rates. This can be seen by evaluating Eq 29 for two different repressor binding sites, labeled (1) and (2), but with the same repressor copy number $R$, and taking the difference to find

$$\Delta F_R^{(1)} - \Delta F_R^{(2)} = \beta \Delta \varepsilon_1 - \beta \Delta \varepsilon_2 = -\log(k_R^+/k_1^-) + \log(k_R^+/k_2^-), \qquad (30)$$

or simply

$$\beta \Delta \varepsilon_1 - \beta \Delta \varepsilon_2 = \log(k_2^-/k_1^-). \qquad (31)$$

The left and right hand sides of this equation are exactly the horizontal and vertical axes of the top panel of Fig 4. Since we inferred rates for three repressor binding sites (O1, O2, and Oid), there are only two independent differences that can be constructed, and we arbitrarily chose to plot O2-O1 and O1-Oid in Fig 4B. Numerically, we compute values of $k_{O1}^-/k_{Oid}^-$ and $k_{O2}^-/k_{O1}^-$ directly from our full posterior samples, which conveniently provides uncertainties as well, as detailed in the S1 Supporting Information. We then compare these log ratios of rates to the binding energy differences $\Delta \varepsilon_{O1} - \Delta \varepsilon_{Oid}$ and from $\Delta \varepsilon_{O2} - \Delta \varepsilon_{O1}$ as computed from the values from both [41] and [43]. Three of the four values are within $\sim 0.5\, k_B T$ of the diagonal representing perfect agreement, which is comparable to the $\sim$ few $\times\, 0.1\, k_B T$ variability between the independent determinations of the same quantities between [41] and [43]. The only outlier involves Oid measurements from [43], and as the authors of [43] note, this is a difficult measurement of low fluorescence signal against high background since Oid represses so strongly. We are therefore inclined to regard the failure of this point to fall near the diagonal as a testament to the difficulty of the measurement and not as a failure of our theory.

On the whole then, we find these to be results very suggestive. Although not conclusively, our results point at the possible interpretation of the phenomenological free energy of transcription factor binding as the log of a ratio of transition rates. This did not need to be the case; the free energy was inferred from bulk measurements of mean gene expression relative to an unregulated strain –the fold-change– while the rates were extracted from fitting single-molecule counts to the full mRNA steady state distribution. Further single molecule experiments along the lines of [55] could shed further light into this idea.

*3.2.3 Comparison with prior measurements of repressor kinetics*

In the previous section we established the equivalence between the equilibrium models' binding energies and the repressor kinetics we infer from mRNA population distributions. But one might worry that the repressor rates we infer from mRNA distributions are *themselves* merely fitting parameters and that they do not actually correspond to the binding and unbinding rates of the repressor in vivo. To verify that this is not the case, we next compare our kinetic rates with a different measurement of the same rates using a radically different method: single molecule measurements as performed in Hammar et. al. [55]. This is plotted in the lower panel of Fig 4B.

Since we do not have access to repressor copy number for either the single-cell mRNA data from [33] or the single-molecule data from [55], we cannot make an apples-to-apples comparison of association rates $k_R^+$. Further, while Hammar et. al. directly measure the dissociation rates $k_R^-$, our inference procedure returns $k_R^-/\gamma$, i.e., the repressor dissociation rate nondimensionalized by the mRNA degradation rate $\gamma$. So to make the comparison, we must make an

assumption for the value of $\gamma$ since it was not directly measured. For most mRNAs in *E. coli*, quoted values for the typical mRNA lifetime $\gamma^{-1}$ range between about 2.5 min [61] to 8 min. We chose $\gamma^{-1} = 3$ min and $\gamma^{-1} = 5$ min as representative values and plot a comparison of $k_{O1}^-$ and $k_{Oid}^-$ from our inference with corresponding values reported in [55] for both these choices of $\gamma$.

The degree of quantitative agreement in the lower panel of Fig 4B clearly depends on the precise choice of $\gamma$. Nevertheless we find this comparison very satisfying, when two wildly different approaches to a measurement of the same quantity yield broadly compatible results. We emphasize the agreement between our rates and the rates reported in [55] for any reasonable $\gamma$: values differ by at most a factor of 2 and possibly agree to within our uncertainties of 10-20%. From this we feel confident asserting that the parameters we have inferred from Jones et. al.'s single-cell mRNA counts data do in fact correspond to repressor binding and unbinding rates, and therefore our conclusions on the agreement of these rates with binding energies from [41] and [43] are valid.

*3.2.4 Model checking*

In Fig 3B we saw that the simple Poisson model of a constitutive promoter, despite having a well behaved posterior, was clearly insufficient to describe the data. It behooves us to carry out a similar check for our model of simple repression, codified by Eq 27 for the steady-state mRNA copy number distribution. As derived in Sections 1 and 2, we have compelling theoretical reasons to believe it is a good model, but if it nevertheless turned out to be badly contradicted by the data we should like to know.

The details are deferred to the S1 Supporting Information, and here we only attempt to summarize the intuitive ideas, as detailed at greater length by Jaynes [62] as well as Gelman and coauthors [51, 63]. From our samples of the posterior distribution, plotted in Fig 4A, we generate many replicate data using a random number generator. In Fig 4C, we plot empirical cumulative distribution functions of the middle 95% quantiles of these replicate data with the actual experimental data from Jones et. al. [33] overlaid, covering all ten experimental conditions spanning repressor binding sites and copy numbers (as well as the constitutive baseline UV5).

The purpose of Fig 4C is simply a graphical, qualitative assessment of the model: do the experimental data systematically disagree with the simulated data, which would suggest that our model is missing important features? A further question is not just whether there is a detectable difference between simulated and experimental data, but whether this difference is likely to materially affect the conclusions we draw from the posterior in Fig 4A. More rigorous and quantitative statistical tests are possible [51], but their quantitativeness does not necessarily make them more useful. As stated in [63], we often find this graphical comparison more enlightening because it better engages our intuition for the model, not merely telling *if* the model is wrong but suggesting *how* the model may be incomplete.

Our broad brush takeaway from Fig 4C is overall of good agreement. There some oddities, in particular the long tails in the data for Oid, 1 ng/mL, and O2, 0.5 ng/mL. The latter is especially odd since it extends beyond the tail of the unregulated UV5 distribution. This is a relatively small number of cells, however, so whether this is a peculiarity of the experimental data, a statistical fluke of small numbers, or a real biological effect is unclear. It is conceivable that there is some very slow timescale switching dynamics that could cause this bimodality, although it is unclear why it would only appear for specific repressor copy numbers. There is also a small offset between experiment and simulation for O2 at the higher repressor copy numbers, especially at 2 and 10 ng/mL. From the estimate of repressor copy numbers from [33], it is possible that the repressor copy numbers here are becoming large enough to partially

invalidate our assumption of a separation of timescales between burst duration and repressor association rate. Another possibility is that the very large number of zero mRNA counts for Oid, 2 ng/mL is skewing its partner datasets through the shared association rate. None of these fairly minor potential caveats cause us to seriously doubt the overall correctness of our model, which further validates its use to compare the thermodynamic models' binding energies to the kinetic models' repressor binding and unbinding rates, as we originally set out to do.

## Discussion

The study of gene expression is one of the dominant themes of modern biology, made all the more urgent by the dizzying pace at which genomes are being sequenced. But there is a troubling Achilles heel buried in all of that genomic data, which is our inability to find and interpret regulatory sequence. In many cases, this is not possible even qualitatively, let alone the possibility of quantitative dissection of the regulatory parts of genomes in a predictive fashion. Other recent work has tackled the challenge of finding and annotating the regulatory part of genomes [2, 27]. Once we have determined the architecture of the regulatory part of the genome, we are then faced with the next class of questions which are sharpened by formulating them in mathematical terms, namely, what are the input-output properties of these regulatory circuits and what knobs control them?

The present work has tackled that question in the context of the first regulatory architecture hypothesized in the molecular biology era, namely, the repressor-operator model of Jacob and Monod [64]. Regulation in that architecture is the result of a competition between a repressor which inhibits transcription and RNAP polymerase which undertakes it. Through the labors of generations of geneticists, molecular biologists and biochemists, an overwhelming amount of information and insight has been garnered into this simple regulatory motif, licensing it as what one might call the "hydrogen atom" of regulatory biology. It is from that perspective that the present paper explores the extent to which some of the different models that have been articulated to describe that motif allow us to understand both the average level of gene expression found in a population of cells, the intrinsic cell-to-cell variability, and the full gene expression distribution found in such a population as would be reported in a single molecule mRNA Fluorescence *in situ* Hybridization experiment, for example. We do so by contrasting two theoretical frameworks to think about the problem. On the one hand, thermodynamic models are convenient ways to describe the problem of gene regulation because of the relatively small number of parameters compared with kinetic models, and the agnostic nature with respect to the full reaction topology that leads to a transcription event. In addition, either implicitly or explicitly, many approaches to gene regulation focus strictly on occupancy and ideas such as $K_d$s, and thus is of great interest to test these models. This comes with the limitation that thermodynamic models are only able to make statements about mean gene expression of a quasi-steady state process. On the other hand, kinetic models break those limitations, being able to make explicit predictions about higher moments of the gene expression distribution, and the transient states towards steady state, at the cost of propagating the number of parameters, and demanding the full set of reactions.

Our key insights can be summarized as follows. First, as shown in Fig 1, the mean expression in the simple repression architecture is captured by a master curve in which the action of repressor and the details of the RNAP interaction with the promoter appear separately and additively in an effective free energy. Interestingly, as has been shown elsewhere in the context of the Monod-Wyman-Changeux model, these kinds of coarse-graining results are an exact mathematical result and do not constitute hopeful approximations or biological naivete [43, 44]. The fact that thermodynamic and kinetic models for the simple repression motif can not

only be reconciled, but are indistinguishable from each other at the level of mean mRNA count is a consequence of the possible transitions between promoter microstates. The promoter microstates in all models shown in Fig 1 can explicitly be separated into two groups: 1) promoters with repressor bound, and 2) all other promoters states. The single path between these two groups of states guarantees detailed-balance between the groups. We can therefore separate *expression* related transitions, which by definition must be out of equilibrium, with *regulation* transitions that can in principle be in equilibrium. This implies that the probability distribution of the promoter microstates not involving the repressor take the same functional form, allowing us to write the fold-change as an effective free energy involving a regulation term, $\Delta F_R$, and a promoter details term, $\rho$. We offer this argument as a conjecture, and we suspect that a careful argument using the graph-theoretic framework based on the Matrix-Tree Theorem [65], might furnish a "proof" not with some challenges as this theoretical framework applies, at the present, to finite graphs.

To further dissect the relative merits of the different models, we must appeal to higher moments of the gene expression probability distribution. To that end, our second set of insights focus on gene expression noise, where it is seen that a treatment of the constitutive promoter already reveals that some models have Fano factors (variance/mean) that are less than one, at odds with any and all experimental data that we are aware of [33, 34]. This theoretical result allows us to directly discard a subset of the models (models 1-3 in Fig 2A) since they cannot be reconciled with experimental observations. The two remaining models (models 4 and 5 in Fig 2) appear to contain enough microscopic realism to be able to reproduce the data. A previous exploration of model 4 demonstrated the "sloppy" [58] nature of the model in which data on single-cell mRNA counts alone cannot constrain the value of all parameters simultaneously [47]. Here we demonstrate that the proposed one-state bursty promoter model (model 5 in Fig 2A) emerges as a limit of the commonly used two-state promoter model [17, 21, 32–34]. We put the idea to the test that this level of coarse-graining is rich enough to reproduce previous experimental observations. In particular we perform Bayesian inference to determine the two parameters describing the full steady-state mRNA distribution, finding that the model is able to provide a quantitative description of a plethora of promoter sequences with different mean levels of expression and noise.

With the results of the constitutive promoter in hand, we then fix the parameters associated with this class of promoters and use them as input for evaluating the noise in gene expression for the simple repression motif itself. This allows us to provide a single overarching analysis of both the constitutive and simple repression architectures using one simple model and corresponding set of self-consistent parameters, demonstrating not only a predictive framework, but also reconciling the thermodynamic and kinetic views of the same simple repression constructs. More specifically, we obtained values for the transcription factor association and dissociation rates by performing Bayesian inference on the full mRNA distribution for data obtained from simple-repression promoters with varying number of transcription factors per cell and affinity of such transcription factors for the binding site. The free energy value obtained from these kinetic rates –computed as the log ratio of the rates– agrees with previous inferences performed only from mean gene expression measurements, that assumed a thermodynamic rather than a kinetic framework [41, 43]. Interestingly, to constrain the binding and unbinding rates inferred for Fig 4, all of the rates need to be simultaneously fit. This is because a single combination of operator (repressor binding site) and aTc concentration (repressor copy number) cannot constrain both rates independently, but only their ratio (See the S1 Supporting Information for further details).

It is important to be clear on the nature of the assumptions present in all models, including model 5 from Fig 2A. First, as with many of the models used routinely in the analysis of

transcription, the models considered here imagine the promoter as existing in a series of discrete states. This assumption is clearly an oversimplification due to effects such as DNA supercoiling [37, 38] or DNA looping [39] which are both parameterized by continuous variables describing DNA shape; although there are formal and very interesting ways of integrating out such degrees of freedom. Another facet of the models used here is that they can feature irreversible processes characterized by only a single rate process with no corresponding return pathway at odds with thermodynamics. Such models are mathematically convenient and serve only as an approximation to the more realistic situation in which the forward rates are much larger than the backward rates. One consequence is that the inferred rates for the transcription burst rate and burst size can only be thought of as effective rate constants. Another idealization of the models considered here is that when computing the fold-change in gene expression (Section 2) we assume that the translation efficiency of an mRNA transcript is the same in cells with and without regulation. There might be some known biological effects that could limit the applicability of this assumption such as cooperation between the transcription and translation machinery [66], or coupling of mRNA transcription and degradation [61] among other potential effects. Another implicit assumption in all of the models discussed here is that the effects due to changes in gene copy number during the cell cycle have no effects on the expression profile. For moderate growth rates such as those used in [33] from which we obtained the data, at all points during the cell cycle there are at most two copies of the gene of interest. In a recent publication, we explored the effects that this change in gene dosage can have in gene expression, finding that the effects are much more drastic at the protein level [47].

More specifically, for model 5 from Fig 2A, it is interesting to speculate what microscopic details are being coarse-grained by our burst rate and burst size in Fig 2A, model 5. Chromosomal locus is one possible influence we have not addressed in this work, as all the single-molecule mRNA data from [33] that we considered was from a construct integrated at the *galK* locus. The results of [37] indicate that transcription-induced supercoiling contributes substantially in driving transcriptional bursting, and furthermore, their Fig 7 suggests that the parameters describing the rate, duration, and size of bursts vary substantially for transcription from different genomic loci. Although the authors of [67] do not address noise, they note enormous differences in mean expression levels when an identical construct is integrated at different genomic loci. The authors of [68] attribute noise and burstiness in their single-molecule mRNA data to the influence of different sigma factors, which is a reasonable conclusion from their data. Could the difference also be due to the different chromosomal locations of the two operons? What features of different loci are and are not important? Could our preferred coarse-grained model capture the variability across different loci? If so, and we were to repeat the parameter inference as done in this work, is there a simple theoretical model we could build to understand the resulting parameters?

In summary, this work took up the challenge of exploring the extent to which a single specific mechanistic model of the simple-repression regulatory architecture suffices to explain the broad sweep of experimental data for this system. Pioneering early experimental efforts from the Müller-Hill lab established the simple-repression motif as an arena for the quantitative dissection of regulatory response in bacteria, with similar work emerging in examples such as the *ara* and *gal* operons as well [28, 29, 69–73]. In light of a new generation of precision measurements on these systems, the definition of what it means to understand them can now be formulated as a rigorous quantitative question. In particular, we believe understanding of the simple repression motif has advanced sufficiently that the design of new versions of the architecture is now possible, based upon predictions about how repressor copy number and DNA binding site strength control expression. In our view, the next step in the progression is to first perform similar rigorous analyses of the fundamental "basis set" of regulatory architectures.

Natural follow-ups to this work are explorations of motifs such as simple activation that is regulated by a single activator binding site, and the repressor-activator architecture, mediated by the binding of both a single activator and a single repressor, and beyond. With the individual input-output functions in hand, similar quantitative dissections including the rigorous analysis of their tuning parameters can be undertaken for the "basis set" of full gene-regulatory networks such as switches, feed-forward architectures and oscillators for example, building upon the recent impressive efforts from systems biologists and synthetic biologists [74, 75].

## Supporting information

**S1 Supporting Information. Detailed mathematical derivations and extended analysis.**
(PDF)

## Acknowledgments

We thank Rob Brewster for providing the raw single-molecule mRNA FISH data. We thank Justin Bois for his key support with the Bayesian inference section. We would also like to thank Griffin Chure for invaluable feedback on the manuscript.

## Author Contributions

**Conceptualization:** Muir Morrison, Manuel Razo-Mejia, Rob Phillips.

**Formal analysis:** Muir Morrison.

**Funding acquisition:** Rob Phillips.

**Investigation:** Muir Morrison.

**Methodology:** Muir Morrison, Rob Phillips.

**Project administration:** Muir Morrison, Manuel Razo-Mejia, Rob Phillips.

**Software:** Muir Morrison, Manuel Razo-Mejia.

**Supervision:** Muir Morrison, Manuel Razo-Mejia, Rob Phillips.

**Validation:** Muir Morrison, Manuel Razo-Mejia, Rob Phillips.

**Visualization:** Muir Morrison, Manuel Razo-Mejia.

**Writing – original draft:** Muir Morrison, Manuel Razo-Mejia, Rob Phillips.

**Writing – review & editing:** Muir Morrison, Manuel Razo-Mejia, Rob Phillips.

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
