## [Decision Letter · Decision Letter 0]

31 Jul 2020

Dear Prof. Phillips,

Thank you very much for submitting your manuscript "Reconciling kinetic and equilibrium models of bacterial transcription" for consideration at PLOS Computational Biology.

As with all papers reviewed by the journal, your manuscript was reviewed by members of the editorial board and by several independent reviewers. In light of the reviews (below this email), we would like to invite the resubmission of a significantly-revised version that takes into account the reviewers' comments. The reviewers generally agree that this paper represents a substantial contribution to the literature, but have raised issues with the length and structure of the presentation.

Although we are happy to give authors wide latitude in how they present their results and we are not opposed to a lengthy paper, we think that your manuscript should follow the basic structure of an article (see: https://journals.plos.org/ploscompbiol/s/submission-guidelines) and allow the readers with a diverse background to understand the main results. In addition, the key methodological details should be presented in the main body of the paper in a separate section. 

We cannot make a final decision about publication until we have seen the revised manuscript and your response to the reviewers' comments. Your revised manuscript will be sent to reviewers for further evaluation.

Sincerely,

James R. Faeder

Associate Editor

PLOS Computational Biology

Jian Ma

Deputy Editor

PLOS Computational Biology

Reviewer's Responses to Questions

**Comments to the Authors:**

**Reviewer #1: **Review: Morrison et al.

The authors present a detailed survey of simple biophysical models for the kinetics of transcription from a synthetic bacterial promoter. In particular, they highlight the challenges to discriminating between different models based on the available data of single-cell mRNA numbers. While the indistinguishability of models would be unsurprising (perhaps even self-evident) to practitioners of single-cell microbiology, the current work does an admirable job of rigorously demonstrating the problem and proposing some possible avenues towards addressing it. Thus, the manuscript is a welcome addition to the field.

Before publication is considered, however, a number of issues must be addressed. Those relate mainly to strong simplifying assumptions shared by all the models presented, assumptions that are made without a proper discussion of their validity. Making the cow spherical is often productive, but when doing so, the reader must be made aware of that, and the deviation from reality clearly stated.

Specifically:

1) “the promoter is imagined to exist in a discrete set of states of occupancy, with each such state of occupancy accorded its own rate of transcription” (page 3). Is there direct experimental evidence for the existence of finite, discrete, activity states? It is easy to imagine how the known biology of bacterial transcription would violate that: For example, the supercoiling state of DNA is a continuous variable, which both modulates transcription and is affected by it (e.g., PMID: 31539491, PMID: 28707908). Its proper consideration would thus fall outside the premise of finite and discrete promoter states.

2) “assuming that the translation efficiency, i.e., the number of proteins translated per mRNA, is the same in both conditions” (page 5). This, too, appears violated by the known biology: The rates of transcription initiation, elongation, translation and degradation are tightly coupled (e.g., PMID: 20413502, PMID: 25964259, PMID: 31539491), hence the ratio of initiation rates is not expected a-priori to map linearly to protein concentration. Is there experimental evidence supporting this assumption? Note that the same argument also holds for measuring the total number of mRNA molecules per cell. To use mRNA number to deduce initiation rates requires showing a linear mapping between the two observables, despite the known coupling of mRNA initiation to elongation and degradation.

3) Another assumption shared by all models presented, but not stated explicitly, is that transcription propensity (per copy) does not change along the cell cycle. This seems especially dubious when modeling a promoter regulated by a repressor, for which it widely believed that transcription is tightly coupled to the passage of the replication fork (PMID: 8526893, PMID: 24562187, PMID: 31527794). How is this assumption justified?

Additional matters:

4) “disproving the universal noise curve from So et. al. [59]” (page 18). How was the universal noise curve disproved? Universal noise/mean relations, similar to those reported in ref #59, were repeatedly observed, in both bacteria and eukaryotes (see, e.g., PMID: 24311680, PMID: 25858977). Moreover, later reports showed that the trend is observed even when accounting for gene dosage effects (PMID: 26965629). Thus, the statement above requires clarification.

5) The manuscript is too long. It reads more like a monograph or a student’s thesis (and a good one, as such) rather than a journal research paper. It is up to the editor, but I would advise that some of the specific models be moved to the supplementary material, to create a more compact text.

**Reviewer #2: **This manuscript examines a few simple linear models of state dependent gene expression used to describe the regulation of gene transcription under control of varying levels of a repressor. The manuscript begins by examining the equilibrium mean level of gene expression, which boils down to expected value of the transcription rate (over the probability mass for the different states) divided by the degradation rate. The authors show that the mean fold change (expression with ‘R’ repressor concentration relative to expression in the absence of the repressor) versus ‘R’ collapses to a simple monotonic function parametrized by (i) an effective free energy term that depends on ‘R’ and (ii) a constant ‘rho’ that depends on the model structure, but not on ‘R’ itself. For non-equilibrium models, a similar form is derived, but where the dependence on ‘R’ is lumped into rate constants k_R^+ and k_R^- for repressor binding/unbinding (presumably with k_R^+ being linearly dependent on the repressor concentration). Next, the manuscript examines the steady state variance predicted by their models, again using a very simple and well-studied class of models for which all propensity functions are linear and where there are no mechanisms of feedback. Then, the manuscript gets into more interesting work with Bayesian parameter inference for the parameters of a simple bursting kinetics model from single-molecule RNA FISH data. They examine published data (from the same group) with nine different combinations of repressor binding sites and repressor concentrations, and they estimate uncertainties in the shared parameters among these nine conditions. The authors argue that this analysis of the previous data confirms the validity of their equilibrium models.

The manuscript addresses an interesting topic of how to connect different types of gene expression models and what mechanistic properties can be inferred from single-cell RNA FISH data, which are both important topics in the single-cell research community. The Bayesian analysis of a single model to describe multiple different combinations of promoters and repressor concentrations is a very nice example for how simple models can be effectively integrated with single-cell data. The results and approaches appear to be correct. Unfortunately, much of the presented model development and analyses (especially sections 2 and 3) was wasted rehashing several extremely well-studied models, so that it became very difficult to discern what new insight is being provided by the authors, and what is a simple restating of known results. This issue was compounded by the fact that the manuscript does not provide adequate citations to existing research in the field, especially given the vast amount of recent work that has been performed on the identification or, and uncertainty quantification for, gene regulation models using single-cell RNA FISH data. Finally, the manuscript makes a big deal about how the authors’ analyses confirm the validity of their equilibrium model, but (1) at least as far as I could find, the authors provide no convincing statistical evidence that supports this claim while rejecting some competing null model and (2) the actual importance or novelty of this claim is not clear. Overall, as written, I expect this manuscript will have a low to moderate impact among the general readership of PLoS Computational Biology, and it might be more appropriate for a more specialized physics or physical biology journal.

Major Comments:

1) The authors first result (Sec. 2, the collapse of the fold change curves for the presented simple linear models) is not surprising for either the equilibrium or non-equilibrium models. Each model described in the first part of the manuscript can be split into two groups of states: (i) a single repressor-bound state and (ii) a bunch of interconnected unbound states. At least for the models described in Fig 1, the authors are aware that there is only a single path to or from the repressor bound state, which guarantees detailed balance between these two groups of states (this would still be true even if the studied models were extended to include multiple different bound repressor states). As such, the conditional steady state probability mass within each group of states will be identical with or without the single repressor state, and this will enable the form shown in Figure 1D — although with multiple bound repressor states, the form for \\Delta F_R could be more complicated with its form dictated by the interactions between the states in that group. Given that the authors only consider one model for the states within the repressor group (i.e., a single bound state), it is not clear what we learn by the observation that all these models have the same functional form for the fold change.

2) In the second part of the manuscript (Sec 3), the authors extend their analysis to explore the variance, and Fano factors, under various simple linear models with multiple expression states. Like all models with linear propensity functions, the first and second moments for these models are known (not just for their steady state values as presented in this manuscript, but also for their temporal correlations and transient responses). The bursting model at the end of section 3 is also well studied in the literature. It was difficult for me to connect the insight from section 2 to that of section 3? Is there any relevance of the fold change functional form from Section 2 when examined through the lens of Fano factors or distributions? Or are these meant to be two unconnected analyses?

3) The use of Bayesian analysis and MCMC to infer parameters from single-cell RNA FISH data is a little more interesting. Although this kind of analysis has now also been done many times in the recent literature, the authors’ application of the MCMC to estimate the posterior parameters under many different promoters (Fig. 3C) was interesting. Again, it is not clear how this section relates back to the Section 2. Perhaps this was meant to be addressed in Fig. 3(D), but I could only find one sentence in the manuscript about this panel (page 29) and it did not provide much insight. Overall, if the focus of the paper could have been on this analysis of multiple data sets, perhaps the paper would have been more compelling to me.

4) The discussion in Sections 4.2.2 and 4.2.3 need to be fleshed out a little more — as written, these sections are very confusing, and it is not clear how one should evaluate the statistical significance of the authors’ claim that this is a “striking confirmation of the validity of the equilibrium models (page 34)”. Can the authors introduce and compare to some sort of null model or provide some simulated data to demonstrate that if their theory of equilibrium models and repressor binding energies were incorrect this analysis would have provided a clear rejection of that theory? Can the authors give a p-value for the term 'striking'?

5) After the very nice parameter estimation shown in Fig. 4(A), I was expecting the model to be tested on its ability to predict the distributions for the held out cases (e.g., Oid at 0.5 or 10 ng/ml or O1 at 0.5 ng/ml) using the parameters inferred from the other promoter/repressor conditions. These predictions would have made for a much more compelling version of Fig. 4(C). As presented, Fig 4(C) only seems to show plots for data that were already used in the fitting.

6) The manuscript is missing important citations to key work in this research domain. Admittedly, there are hundreds (if not thousands) of theoretical and experimental papers that invoke the bursting gene expression models, a comprehensive list would be impossible. But since the authors saw fit to cite R. Phillips’ work at least 18 times, they could have made a little more effort to capture the other relevant work on the topic. Some particularly relevant articles would include work by Hana El Samad, Ramon Grima, Srividya Iyer-Biswas, Michal Komorowski, Mustafa Khammash, Heinz Koeppl, Andrew Mugler, Brian Munsky, Arjun Raj, Jakub Reuss, Abhyudai Singh, Eduardo Sontag, Michael Stumpf, and Alexander van Oudenaarden, to list just a handful of research labs that have presented extensive analyses of bursting gene expression models, have fit these models to single-cell data (including RNA FISH data), have quantified parameter uncertainties using MCMC analyses, and more.

Minor Comments.

7) Please introduce the variable ‘R” and other necessary notation before the first reference to Figure 1. It is confusing to use both \\beta and k_bT in the figure. Please choose one or the other. Similarly, the variable N_{NS} is used in Fig 1 too long before it is defined under Eqn 4. This is confusing.

8) In Fig. 4C, the use of ECDF is nice for distributions whose support is well spread over the positive integer numbers (e.g., for the low repressor concentrations), but this visualization can hide deviations for distributions whose support is mostly at zero (e.g., for high repressor concentrations). It would be useful for the authors to show the histograms and probability mass vectors as well as the ECDFs. The authors are aware of these differences, and it is much appreciated that they discuss them in Section 4.2.4, but I worry that these ‘oddities’ or ‘minor quibbles’ are inadvertently diminished by the choice to use ECDFs instead of probability density histograms.

9) Typo on page 32: “for a given inducer concentrations.“ Remove ’s’ from ‘concentrations’

10) While this may be a matter of personal taste, I found the informal conversational tone of the manuscript (e.g., words like ‘beautiful’, ‘bonanza’, 'quibble', etc) to be distracting. Such flowery language is great for a short perspectives article or opinion piece, but for a manuscript of this length, rigor is key and imprecise language becomes annoying. It would be nice if the authors could tighten up their message and try to present a more concise description of their key findings.

**Reviewer #3: **Please see uploaded comments.

**Reviewer #4:** Review is uploaded as an attachment.

**Have all data underlying the figures and results presented in the manuscript been provided?**

Reviewer #1: Yes

Reviewer #2: Yes

Reviewer #3: Yes

Reviewer #4: Yes

PLOS authors have the option to publish the peer review history of their article (what does this mean?). If published, this will include your full peer review and any attached files.

Reviewer #1: No

Reviewer #2: **Yes: **Brian Munsky

Reviewer #3: **Yes: **Jeremy Gunawardena

Reviewer #4: **Yes: **Michael E Wall
---

## [Decision Letter · Decision Letter 1]

28 Nov 2020

Dear Prof. Phillips,

We are pleased to inform you that your manuscript 'Reconciling kinetic and thermodynamic models of bacterial transcription' has been provisionally accepted for publication in PLOS Computational Biology.

Before your manuscript can be formally accepted you will need to complete some formatting changes, which you will receive in a follow up email. In addition, at this stage please feel free to address in the revisions at your discretion the comments of reviewer 2. 

A member of our team will be in touch with a set of requests.

Best regards,

James R. Faeder

Associate Editor

PLOS Computational Biology

Jian Ma

Deputy Editor

PLOS Computational Biology

Reviewer's Responses to Questions

**Comments to the Authors:**

Reviewer #1: The authors have addressed my concerns.

Reviewer #2: Review is uploaded as an attachment.

Reviewer #3: Please see uploaded file.

Reviewer #4: Based on the response to reviewers, the authors appear to have adequately addressed my previous comments.

In my original review, there was a typo in the following comment, corrected here:

p. 26, Eq. (51). See Paulsson and Ehrenberg, 2000 (https://journals.aps.org/ prl/pdf/10.1103/PhysRevLett.84.5447). Also see Friedman et al, 2006 (https: //doi.org/10.1103/PhysRevLett.97.168302) which derives a continuous limit.

The original review referred to Eq. (52) instead. I was pointing out that similar equations had been derived previously in the literature. I hope this helps clarify why these references were cited.

**Have all data underlying the figures and results presented in the manuscript been provided?**

Reviewer #1: Yes

Reviewer #2: Yes

Reviewer #3: Yes

Reviewer #4: Yes

PLOS authors have the option to publish the peer review history of their article (what does this mean?). If published, this will include your full peer review and any attached files.

Reviewer #1: No

Reviewer #2: **Yes: **Brian Munsky

Reviewer #3: **Yes: **Jeremy Gunawardena

Reviewer #4: **Yes: **Michael Wall

---

## [Editor Report · Acceptance letter]

11 Jan 2021

PCOMPBIOL-D-20-01024R1 

Reconciling kinetic and thermodynamic models of bacterial transcription

Dear Dr Phillips,

I am pleased to inform you that your manuscript has been formally accepted for publication in PLOS Computational Biology. Your manuscript is now with our production department and you will be notified of the publication date in due course.

With kind regards,

Jutka Oroszlan
